# Tree-Structured Recurrent Switching Linear Dynamical Systems for Multi-Scale Modeling

**Josue Nassar**
Department of Electrical & Computer Engineering
Stony Brook University
Stony Brook, NY 11794
`josue.nassar@stonybrook.edu`

**Scott W. Linderman**
Department of Statistics
Columbia University
New York, NY 10027
`scott.linderman@columbia.edu`

**Mónica F. Bugallo**
Department of Electrical & Computer Engineering
Stony Brook University
Stony Brook, NY, 11794
`monica.bugallo@stonybrook.edu`

**Il Memming Park**
Department of Neurobiology and Behavior
Stony Brook University
Stony Brook, NY, 11794
`memming.park@stonybrook.edu`

## Abstract

Many real-world systems studied are governed by complex, nonlinear dynamics. By modeling these dynamics, we can gain insight into how these systems work, make predictions about how they will behave, and develop strategies for controlling them. While there are many methods for modeling nonlinear dynamical systems, existing techniques face a trade off between offering interpretable descriptions and making accurate predictions. Here, we develop a class of models that aims to achieve both simultaneously, smoothly interpolating between simple descriptions and more complex, yet also more accurate models[1]. Our probabilistic model achieves this multi-scale property through a hierarchy of locally linear dynamics that jointly approximate global nonlinear dynamics. We call it the tree-structured recurrent switching linear dynamical system. To fit this model, we present a fully-Bayesian sampling procedure using Pólya-Gamma data augmentation to allow for fast and conjugate Gibbs sampling. Through a variety of synthetic and real examples, we show how these models outperform existing methods in both interpretability and predictive capability.

## 1 Introduction

Complex systems can often be described at multiple levels of abstraction. A computer program can be characterized by the list of functions it calls, the sequence of statements it executes, or the assembly instructions it sends to the microprocessor. As we zoom in, we gain an increasingly nuanced view of the system and its dynamics. The same is true of many natural systems. For example, brain activity can be described in terms of high-level psychological states or via detailed ion channel activations; different tasks demand different levels of granularity. One of our principal aims as scientists is to identify appropriate levels of abstraction for complex natural phenomena and to discover the dynamics that govern how these systems behave at each level of resolution.

Modern machine learning offers a powerful toolkit to aid in modeling the dynamics of complex systems. Bayesian state space models and inference algorithms enable posterior inference of the latent states of a system and the parameters that govern their dynamics (Särkkä, 2013; Barber et al., 2011; Doucet et al., 2001). In recent years, this toolkit has been expanded to incorporate increasingly flexible components like Gaussian processes (Frigola et al., 2014) and neural networks (Chung et al., 2015; Johnson et al., 2016; Gao et al., 2016; Krishnan et al., 2017) into probabilistic time series models. In

---

[1]This work was supported by National Science Foundation (NSF IIS-1734910, CCF-1617986, HRD-1612689) and National Institute of Health (NIH R01EB026946). SWL was supported by the Simons Collaboration on the Global Brain (SCGB-418011).

neuroscience, sequential autoencoders offer highly accurate models of brain activity (Pandarinath et al., 2018). However, while these methods offer state of the art predictive models, their dynamics are specified at only the most granular resolution, leaving the practitioner to tease out higher level structure post hoc.

Here we propose a probabilistic generative model that provides a multi-scale view of the dynamics through a hierarchical architecture. We call it the *tree-structured recurrent switching linear dynamical system*, or TrSLDS. The model builds on the recurrent SLDS (Linderman et al., 2017) to approximate latent nonlinear dynamics through a hierarchy of locally linear dynamics. Once fit, the TrSLDS can be queried at different levels of the hierarchy to obtain dynamical descriptions at multiple levels of resolution. As we proceed down the tree, we obtain higher fidelity, yet increasingly complex, descriptions. Thus, depth offers a simple knob for trading off interpretability and flexibility. The key contributions are two-fold[2]: first, we introduce a new form of tree-structured stick breaking for multinomial models that strictly generalizes the sequential stick breaking of the original rSLDS, while still permitting Pólya-gamma data augmentation (Polson et al., 2013) for efficient posterior inference; second, we develop a hierarchical prior that links dynamics parameters across levels of the tree, thereby providing descriptions that vary smoothly with depth. The paper is organized as follows. Section 2 provides background material on switching linear dynamical systems and their recurrent variants. Section 3 presents our tree-structured model and Section 4 derives an efficient fully-Bayesian inference algorithm for the latent states and dynamics parameters. Finally, in Section 5 we show how our model yields multi-scale dynamics descriptions for synthetic data from two standard nonlinear dynamical systems—the Lorenz attractor and the FitzHugh-Nagumo model of nonlinear oscillation—as well as for a real dataset of neural responses to visual stimuli in a macaque monkey.

## 2    BACKGROUND

Let $x_t \in \mathbb{R}^{d_x}$ and $y_t \in \mathbb{R}^{d_y}$ denote the latent state and the observation of the system at time $t$ respectively. The system can be described using a state-space model:

$$x_t = f(x_{t-1}, w_t; \Theta), \quad w_t \sim \mathrm{F}_w \quad \textit{(state dynamics)} \tag{1}$$

$$y_t = g(x_t, v_t; \Psi), \quad\quad v_t \sim \mathrm{F}_v \quad\quad \textit{(observation)} \tag{2}$$

where $\Theta$ denotes the dynamics parameters, $\Psi$ denotes the emission (observation) parameters, and $w_t$ and $v_t$ are the state and observation noises respectively. For simplicity, we restrict ourselves to systems of the form:

$$x_t = f(x_{t-1}; \Theta) + w_t, \quad w_t \sim \mathcal{N}(0, Q), \tag{3}$$

If the state space model is completely specified then recursive Bayesian inference can be applied to obtain an estimate of the latent states using the posterior $p\left(x_{0:T} | y_{1:T}\right)$ (Doucet et al., 2001). However in many applications, the parametric form of the state space model is unknown. While there exist methods that perform smoothing to obtain an estimate of $x_{0:T}$ (Barber, 2006; Fox et al., 2009; Djuric & Bugallo, 2006), we are often interested in not only obtaining an estimate of the continuous latent states but also in learning the dynamics $f(\cdot; \Theta)$ that govern the dynamics of the system.

In the simplest case, we can take a parametric approach to solving this joint state-parameter estimation problem. When $f(\cdot; \Theta)$ and $g(\cdot; \Psi)$ are assumed to be linear functions, the posterior distribution over latent states is available in closed-form and the parameters can be learned via expectation-maximization. On the other hand, we have nonparametric methods that use Gaussian processes and neural networks to learn highly nonlinear dynamics and observations where the joint estimation is untractable and approximations are necessarily imployed (Zhao & Park, 2016; 2018; Frigola et al., 2014; Sussillo et al., 2016). Switching linear dynamical systems (SLDS) (Ackerson & Fu, 1970; Chang & Athans, 1978; Hamilton, 1990; Ghahramani & Hinton, 1996; Murphy, 1998) balance between these two extremes, approximating the dynamics by stochastically transitioning between a small number of linear regimes.

---

[2]Source code is available at `https://github.com/catniplab/tree_structured_rslds`

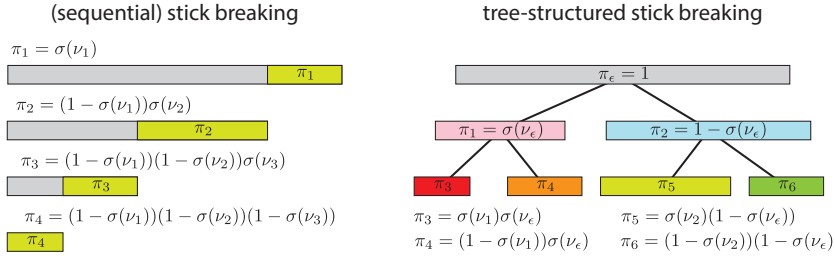

Figure 1: State probability allocation through stick-breaking in standard rSLDS and the TrSLDS.

## 2.1 SWITCHING LINEAR DYNAMICAL SYSTEMS

SLDS approximate nonlinear dynamics by switching between a discrete set of linear regimes. An additional discrete latent state $z_t \in \{1, \dots, K\}$ determines the linear dynamics at time $t$,

$$x_t = x_{t-1} + A_{z_t}x_{t-1} + b_{z_t} + w_t, \quad w_t \sim \mathcal{N}(0, Q_{z_t}) \tag{4}$$

where $A_k, Q_k \in \mathbb{R}^{d_x \times d_x}$ and $b_k \in \mathbb{R}^{d_x}$ for $k = 1, \dots, K$. Typically, $z_t$ is endowed with Markovian dynamics, $\Pr(z_t|z_{t-1} = k) = \boldsymbol{\pi_k}$. The conditionally linear dynamics allow for fast and efficient learning of the model and can utilize the learning tools developed for linear systems (Haykin, 2001). While SLDS can estimate the continuous latent states $x_{0:T}$, the assumption of Markovian dynamics for the discrete latent states severely limits their generative capacity.

## 2.2 RECURRENT SWITCHING LINEAR DYNAMICAL SYSTEMS

Recurrent switching linear dynamical systems (rSLDS) (Linderman et al., 2017), also known as augmented SLDS (Barber, 2006), are an extension of SLDS where the transition density of the discrete latent state depends on the previous location in the continuous latent space

$$z_t|x_{t-1}, \{R, r\} \sim \pi_{SB}(\nu_t), \tag{5}$$
$$\nu_t = Rx_{t-1} + r, \tag{6}$$

where $R \in \mathbb{R}^{K-1 \times d_x}$ and $r \in \mathbb{R}^{K-1}$ represents hyperplanes. $\pi_{SB} : \mathbb{R}^{K-1} \to [0,1]^K$ maps from the reals to the probability simplex via stick-breaking:

$$\pi_{SB}(\nu) = \left(\pi_{SB}^{(1)}(\nu), \cdots, \pi_{SB}^{(K)}(\nu)\right), \quad \pi_{SB}^{(k)} = \sigma(\nu_k) \prod_{j<k} \sigma(-\nu_j), \tag{7}$$

for $k = 1, \dots, K-1$ and $\pi_{SB}^{(K)} = \prod_{k=1}^{K-1} \sigma(-\nu_k)$ where $\nu_k$ is the $k$th component of of $\nu$ and $\sigma(\nu) = (1 + e^{-\nu})^{-1}$ is the logistic function (Fig. 1). By including this recurrence in the transition density of $z_t$, the rSLDS partitions the latent space into $K$ sections, where each section follows its own linear dynamics. It is through this combination of locally linear dynamical systems that the rSLDS approximates eq. (3); the partitioning of the space allows for a more interpretable visualization of the underlying dynamics.

Recurrent SLDS can be learned efficiently and in a fully Bayesian manner, and experiments empirically show that they are adept in modeling the underlying generative process in many cases. However, the stick breaking process used to partition the space poses problems for inference due to its dependence on the permutation of the discrete states $\{1, \cdots, K\}$ (Linderman et al., 2017).

## 3 TREE-STRUCUTRED RECURRENT SWITCHING LINEAR DYNAMICAL SYSTEMS

Building upon the rSLDS, we propose the tree-structured recurrent switching linear dynamical system (TrSLDS). Rather than sequentially partitioning the latent space using stick breaking, we use a tree-structured stick breaking procedure (Adams et al., 2010) to partition the space.

Let $\mathcal{T}$ denote a tree structure with a finite set of nodes $\{\epsilon, 1, \cdots, N\}$. Each node $n$ has a parent node denoted by $\mathrm{par}(n)$ with the exception of the root node, $\epsilon$, which has no parent. For simplicity, we initially restrict our scope to balanced binary trees where every internal node $n$ is the parent of two children, $\mathrm{left}(n)$ and $\mathrm{right}(n)$. Let $\mathrm{child}(n) = \{\mathrm{left}(n), \mathrm{right}(n)\}$ denote the set of children for internal node $n$. Let $\mathcal{Z} \subseteq \mathcal{T}$ denote the set of leaf nodes, which have no children. Let $\mathrm{depth}(n)$ denote the depth of a node $n$ in the tree, with $\mathrm{depth}(\epsilon) = 0$.

At time instant $t$, the discrete latent state $z_t$ is chosen by starting at the root node and traversing down the tree until one of the $K$ leaf nodes are reached. The traversal is done through a sequence of left/right choices by the internal nodes. Unlike in standard regression trees where the choices are deterministic (Lakshminarayanan, 2016), we model the choices as random variables. The traversal through the tree can be described as a stick breaking process. We start at the root node with a unit-length stick $\pi_\epsilon = 1$, which we divide between its two children. The left child receives a fraction $\pi_{\mathrm{left}(\epsilon)} = \sigma(\nu_\epsilon)$ and the right child receives the remainder $\pi_{\mathrm{right}(\epsilon)} = 1 - \sigma(\nu_\epsilon)$ such that $\nu_\epsilon \in \mathbb{R}$ specifies the left/right balance. This process is repeated recursively, subdividing $\pi_n$ into two pieces at each internal node until we reach the leaves of the tree (Fig. 1). The stick assigned to each node is thus,

$$\pi_n = \begin{cases} \sigma(\nu_{\mathrm{par}(n)})^{\mathbb{I}[n=\mathrm{left}(\mathrm{par}(n))]} \left(1 - \sigma(\nu_{\mathrm{par}(n)})\right)^{\mathbb{I}[n=\mathrm{right}(\mathrm{par}(n))]} \pi_{\mathrm{par}(n)} & n \neq \epsilon, \\ 1 & n = \epsilon. \end{cases} \tag{8}$$

We incorporate this into the TrSLDS by allowing $\nu_n$ to be a function of the continuous latent state

$$\nu_n(x_{t-1}, R_n, r_n) = R_n^T x_{t-1} + r_n, \tag{9}$$

where the parameters $R_n$ and $r_n$ specify a linear hyperplane in the continuous latent state space. As the continuous latent state $x_{t-1}$ evolves, the left/right choices become more or less probable. This in turn changes the probability distribution $\pi_k(x_{t-1}, \Gamma, \mathcal{T})$ over the $K$ leaf nodes, where $\Gamma = \{R_n, r_n\}_{n \in \mathcal{T}}$. In the TrSLDS, these leaf nodes correspond to the discrete latent states of the model, such that for each leaf node $k$,

$$p(z_t = k \mid x_{t-1}, \Gamma, \mathcal{T}) = \pi_k(x_{t-1}, \Gamma, \mathcal{T}). \tag{10}$$

In general, the tree-structured stick-breaking is not restricted to balanced binary trees. We can allow more than two children through an ordered sequential stick-breaking at each level. In this sense, tree-structured stick-breaking is a strict generalization of stick-breaking. We also note that similar to rSLDS, the model can be made more flexible by introducing a dependence on the previous discrete latent in eq. (9) but for the rest of the paper, we stick to eq. (8).

## 3.1 A Hierarchical Dynamics Prior that Respects the Tree Structure

Similar to standard rSLDS, the dynamics are conditionally linear given a leaf node $z_t$. A priori, it is natural to expect that locally linear dynamics of nearby regions in the latent space are similar. Thus, in the context of tree-structured stick breaking, we impose that partitions that share a common parent should have similar dynamics. We explicitly model this by enforcing a hierarchical prior on the dynamics that respects the tree structure.

Let $\{A_n, b_n\}$ be the dynamics parameters associated with node $n$. Although the locally linear dynamics of a discrete state are specified by the leaf nodes, we introduce dynamics at the internal nodes as well. These internal dynamics serve as a link between the leaf node dynamics via a hierarchical prior,

$$\mathrm{vec}([A_n, b_n]) | \, \mathrm{vec}([A_{\mathrm{par}(n)}, b_{\mathrm{par}(n)}]) \sim \mathcal{N}(\mathrm{vec}([A_{\mathrm{par}(n)}, b_{\mathrm{par}(n)}]), \Sigma_n), \tag{11}$$

where $\mathrm{vec}(\cdot)$ is the vectorization operator. The prior on the root node is

$$\mathrm{vec}([A_\epsilon, b_\epsilon]) \sim \mathcal{N}(0, \Sigma_\epsilon). \tag{12}$$

We impose the following constraint on the covariance matrix of the prior

$$\Sigma_n = \lambda^{\mathrm{depth}(n)} \Sigma_\epsilon, \tag{13}$$

where $\lambda \in (0, 1)$ is a hyper parameter that dictates how "close" a parent and child are to one another. The prior over the parameters can be written as, where the affine term and the $\mathrm{vec}(\cdot)$ operator are dropped for compactness,

$$p(\{A_n\}_{n \in \mathcal{T}}) = p(A_\epsilon) \prod_{i \in \mathrm{child}(\epsilon)} p(A_i | A_\epsilon) \prod_{j \in \mathrm{child}(i)} p(A_j | A_i) \, \ldots \prod_{z \in \mathcal{Z}} p(A_z | A_{\mathrm{par}(z)}). \tag{14}$$

It is through this hierarchical tree-structured prior that TrSLDS obtains a multi-scale view of the system. Parents are given the task of learning a higher level description of the dynamics over a larger region while children are tasked with learning the nuances of the dynamics. The use of hierarchical priors also allows for neighboring sections of latent space to share common underlying dynamics inherited from their parent. TrSLDS can be queried at different levels, where levels deeper in the tree provide more resolution.

TrSLDS shares some features with regression trees (Lakshminarayanan, 2016), even though regression trees are primarily used for standard, static regression problems. The biggest differences are that our tree-structured model has stochastic choices and the internal nodes contribute to smoothing across partitions through the corresponding hierarchical prior.

There are other hierarchical extensions of SLDS that have been proposed in the literature. In Stanculescu et al. (2014), they propose adding a layer to factorized SLDS where the top-level discrete latent variables determine the conditional distribution of $z_t$, with no dependence on $x_{t-1}$. While the tree-structured stick-breaking used in TrSLDS is also a hierarchy of discrete latent variables, the model proposed in Stanculescu et al. (2014) has no hierarchy of dynamics, preventing it from obtaining a multi-scale view of the dynamics. In Zoeter & Heskes (2003), the authors construct a tree of SLDSs where an SLDS with $K$ possible discrete states is first fit. An SLDS with $M$ discrete states is then fit to each of the $K$ clusters of points. This process continues iteratively, building a hierarchical collection of SLDSs that allow for a multi-scale, low-dimensional representation of the observed data. While similar in spirit to TrSLDS, there are key differences between the two models. First, it is through the tree-structured prior that TrSLDS obtains a multi-scale view of the dynamics, thus we only need to fit *one* instantiation of TrSLDS; in contrast, they fit a separate SLDS for each node in the tree, which is computationally expensive. There is also no explicit probabilistic connection between the dynamics of a parent and child in Zoeter & Heskes (2003). We also note that TrSLDS aims to learn a multi-scale view of the *dynamics* while Zoeter & Heskes (2003) focuses on smoothing, that is, they aim to learn a multi-scale view of the *latent states* corresponding to data but not suitable for forecasting.

In the next section we show an alternate view of TrSLDS which we will refer to as the *residual model* in which internal nodes do contribute to the dynamics. Nevertheless, this residual model will turn out to be equivalent to the TrSLDS.

## 3.2 Residual Model

Let $\{\tilde{A}_n, \tilde{b}_n\}$ be the linear dynamics of node $n$ and let $\text{path}(n) = (\epsilon, \dots, n)$ be the sequence of nodes visited to arrive at node $n$. In contrast to TrSLDS, the dynamics for a leaf node are now determined by **all** the nodes in the tree:

$$p(x_t|x_{t-1}, \tilde{\Theta}, z_t) = \mathcal{N}(x_t|x_{t-1} + \bar{A}_{z_t} x_{t-1} + \bar{b}_{z_t}, \tilde{Q}_{z_t}), \tag{15}$$

$$\bar{A}_{z_t} = \sum_{j \in \text{path}(z_t)} \tilde{A}_j, \quad \bar{b}_{z_t} = \sum_{j \in \text{path}(z_t)} \tilde{b}_j, \tag{16}$$

We model the dynamics to be **independent** a priori, where once again the $\text{vec}(\cdot)$ operator and the affine term aren't shown for compactness,

$$p(\{\tilde{A}_n\}_{n \in \mathcal{T}}) = \prod_{n \in \mathcal{T}} p(\tilde{A}_n), \quad p(\tilde{A}_n) = \mathcal{N}(0, \tilde{\Sigma}_n), \tag{17}$$

where $\tilde{\Sigma}_n = \tilde{\lambda}^{\text{depth}(n)} \tilde{\Sigma}_\epsilon$ and $\tilde{\lambda} \in (0, 1)$.

The residual model offers a different perspective of TrSLDS. The covariance matrix can be seen as representing how much of the dynamics a node is tasked with learning. The root node is given the broadest prior because it is present in eq. (16) for all leaf nodes; thus it is given the task of learning the global dynamics. The children then have to learn to explain the residuals of the root node. Nodes deeper in the tree become more associated with certain regions of the space, so they are tasked with learning more localized dynamics which is represented by the prior being more sharply centered on 0. The model ultimately learns a multi-scale view of the dynamics where the root node captures a coarse estimate of the system while lower nodes learn a much finer grained picture. We show that TrSLDS and residual model yield the same joint distribution (See A for the proof).

**Theorem 1.** *TrSLDS and the residual model are equivalent if the following conditions are true:*
$A_\epsilon = \tilde{A}_\epsilon$, $A_n = \sum_{j \in \text{path}(n)} \tilde{A}_j$, $Q_z = \tilde{Q}_z \ \forall z \in \text{leaves}(\mathcal{T})$, $\Sigma_\epsilon = \tilde{\Sigma}_\epsilon$ *and* $\lambda = \tilde{\lambda}$

## 4 BAYESIAN INFERENCE

The linear dynamic matrices $\Theta$, the hyperplanes $\Gamma = \{R_n, r_n\}_{n \in \mathcal{T} \backslash \mathcal{Z}}$, the emission parameters $\Psi$, the continuous latent states $x_{0:T}$ and the discrete latent states $z_{1:T}$ must be inferred from the data. Under the Bayesian framework, this implies computing the posterior,

$$p(x_{0:T}, z_{0:T}, \Theta, \Psi, \Gamma | y_{1:T}) = \frac{p(x_{0:T}, z_{1:T}, \Theta, \Psi, \Gamma, y_{1:T})}{p(y_{1:T})}. \tag{18}$$

We perform fully Bayesian inference via Gibbs sampling (Brooks et al., 2011) to obtain samples from the posterior distribution described in eq. (18). To allow for fast and closed form conditional posteriors, we augment the model with Pólya-gamma auxiliary variables Polson et al. (2013).

### 4.1 PÓLYA-GAMMA AUGMENTATION

Consider a logistic regression from regressor $x_n \in \mathbb{R}^{d_x}$ to categorical distribution $z_n \in \{0, 1\}$; the likelihood is

$$p(z_{1:N}) = \prod_{n=1}^{N} \frac{\left(e^{x_n^T \beta}\right)^{z_n}}{1 + e^{x_n^T \beta}}. \tag{19}$$

If a Gaussian prior is placed on $\beta$ then the model is non-conjugate and the posterior can't be obtained in closed form. To circumvent this problem Polson et al. (2013) introduced a Pólya-Gamma (PG) augmentation scheme. This augmentation scheme is based on the following integral identity

$$\frac{\left(e^{\psi}\right)^a}{(1 + e^{\psi})^b} = 2^{-b} e^{\kappa \psi} \int_0^{\infty} e^{-\frac{1}{2} \omega \psi^2} p(\omega) \mathrm{d}\omega \tag{20}$$

where $\kappa = a - b/2$ and $\omega \sim \mathrm{PG}(b, 0)$. Setting $\psi = x^T \beta$, it is evident that the integrand is a kernel for a Gaussian. Augmenting the model with PG axillary r.v.s $\{\omega_n\}_{n=1}^N$, eq. (19) can be expressed as

$$p(z_{1:N}) = \prod_{n=1}^{N} \frac{\left(e^{x_n^T \beta}\right)^{z_n}}{1 + e^{x_n^T \beta}} \propto \prod_{n=1}^{N} e^{\kappa_n \psi_n} \int_0^{\infty} e^{-\frac{1}{2} \omega_n \psi_n^2} p(\omega_n) \mathrm{d}\omega_n = \prod_{n=1}^{N} \mathbb{E}_{\omega_n} [e^{-\frac{1}{2}(\omega_n \psi_n^2 - 2\kappa_n \psi_n)}]. \tag{21}$$

Conditioning on $\omega_n$, the posterior of $\beta$ is

$$p(\beta | \omega_{1:N}, z_{1:N}, x_{1:N}) \propto p(\beta) \prod_{n=1}^{N} e^{-\frac{1}{2}(\omega_n \psi_n^2 - 2\kappa_n \psi_n)} \tag{22}$$

where $\psi_n = x_n^T \beta$ and $\kappa_n = z_n - \frac{1}{2}$. It can be shown that the conditional posterior of $\omega_n$ is also PG where $\omega_n | \beta, x_n, z_n \sim \mathrm{PG}(1, \psi_n)$ (Polson et al., 2013).

### 4.2 CONDITIONAL POSTERIORS

The structure of the model allows for closed form conditional posterior distributions that are easy to sample from. For clarity, the conditional posterior distributions for the TrSLDS are given below:

1. The linear dynamic parameters $(A_k, b_k)$ and state variance $Q_k$ of a leaf node $k$ are conjugate with a Matrix Normal Inverse Wishart (MNIW) prior

$$p((A_k, b_k), Q_k | x_{0:T}, z_{1:T}) \propto p((A_k, b_k), Q_k) \prod_{t=1}^{T} \mathcal{N}(x_t | x_{t-1} + A_{z_t} x_{t-1} + b_{z_t}, Q_{z_t})^{\mathbb{1}[z_t = k]}.$$

2. The linear dynamic parameters of an internal node $n$ are conditionally Gaussian given a Gaussian prior on $(A_n, b_n)$

$$p((A_n, b_n) | \Theta_{-n}) \propto p((A_n, b_n) | (A_{\mathrm{par}(n)}, b_{\mathrm{par}(n)})) \prod_{j \in \mathrm{child}(n)} p((A_j, b_j) | (A_n, b_n)).$$

3. If we assume the observation model is linear and with additive white Gaussian noise then the emission parameters $\Psi = \{(C, d), S\}$ are also conjugate with a MNIW prior

$$p((C, d), S|x_{1:T}, y_{1:T}) \propto p((C, d), S) \prod_{t=1}^{T} \mathcal{N}(y_t|Cx_t + d, S).$$

We can also handle Bernoulli observations through the use of Pólya-gamma augmentation. In the interest of space, the details are explained in Section B.1 in the Appendix.

4. The choice parameters are logistic regressions which follow from the conditional posterior

$$p(\Gamma|x_{0:T}, z_{1:T}) \propto p(\Gamma) \prod_{t=1}^{T} p(z_t|x_{t-1}, \Gamma) = p(\Gamma) \prod_{t=1}^{T} \prod_{n \in \text{path}(z_t) \backslash \mathcal{Z}} \frac{(e^{\nu_{n,t}})^{\mathbb{1}(\text{left}(n) \in \text{path}(z_t))}}{1 + e^{\nu_{n,t}}},$$

where $\nu_{n,t} = R_n^T x_{t-1} + r_n$. The likelihood is of the same form as the left hand side of eq. (20), thus it is amenable to the PG augmentation. Let $\omega_{n,t}$ be the auxiliary Pólya-gamma random variable introduced at time $t$ for an internal node $n$. We can express the posterior over the hyperplane of an internal node $n$ as:

$$p((R_n, r_n)|x_{0:T}, z_{1:T}, \omega_{n,1:T}) \propto p((R_n, r_n)) \prod_{t=1}^{T} \mathcal{N}(\nu_{n,t}|\kappa_{n,t}/\omega_{n,t}, 1/\omega_{n,t})^{\mathbb{1}(n \in \text{path}(z_t))},$$

(23)

where $\kappa_{n,t} = \frac{1}{2}\mathbb{1}[j = \text{left}(n)] - \frac{1}{2}\mathbb{1}[j = \text{right}(n)]$, $j \in \text{child}(n)$. Augmenting the model with Pólya-gamma random variables allows for the posterior to be conditionally Gaussian under a Gaussian prior.

5. Conditioned on the discrete latent states, the continuous latent states are Gaussian. However, the presence of the tree-structured recurrence potentials $\psi(x_{t-1}, z_t)$ introduced by eq. (10) destroys the Gaussinity of the conditional. When the model is augmented with PG random variables $\omega_{n,t}$, the augmented recurrence potential, $\psi(x_{t-1}, z_t, \omega_{n,t})$, becomes effectively Gaussian, allowing for the use of message passing for efficient sampling. Linderman et al. (2017) shows how to perform message-passing using the Pólya-gamma augmented recurrence potentials $\psi(x_t, z_t, w_{n,t})$. In the interest of space, the details are explained in Section B.2 in the Appendix.

6. The discrete latent variables $z_{1:T}$ are conditionally independent given $x_{1:T}$ thus

$$p(z_t = k|x_{1:T}, \Theta, \Gamma) = \frac{p(x_t|x_{t-1}, \theta_k) p(z_t = k|x_{t-1}, \Gamma)}{\sum_{l \in \text{leaves}(\mathcal{T})} p(x_t|x_{t-1}, \theta_l) p(z_t = l|x_{t-1}, \Gamma)}, \quad k \in \text{leaves}(\mathcal{T}).$$

7. The conditional posterior of the Pólya-Gamma random variables are also Pólya-Gamma: $\omega_{n,t}|z_t, (R_n, r_n), x_{t-1} \sim \text{PG}(1, \nu_{n,t})$.

Due to the complexity of the model, good initialization is critical for the Gibbs sampler to converge to a mode in a reasonable number of iterations. Details of the initialization procedure are contained in Section C in the Appendix.

## 5 Experiments

We demonstrate the potential of the proposed model by testing it on a number of non-linear dynamical systems. The first, FitzHugh-Nagumo, is a common nonlinear system utilized throughout neuroscience to describe an action potential. We show that the proposed method can offer different angles of the system. We also compare our model with other approaches and show that we can achieve state of the art performance. We then move on to the Lorenz attractor, a chaotic nonlinear dynamical system, and show that the proposed model can once again break down the dynamics and offer an interesting perspective. Finally, we apply the proposed method on the data from Graf et al. (2011).

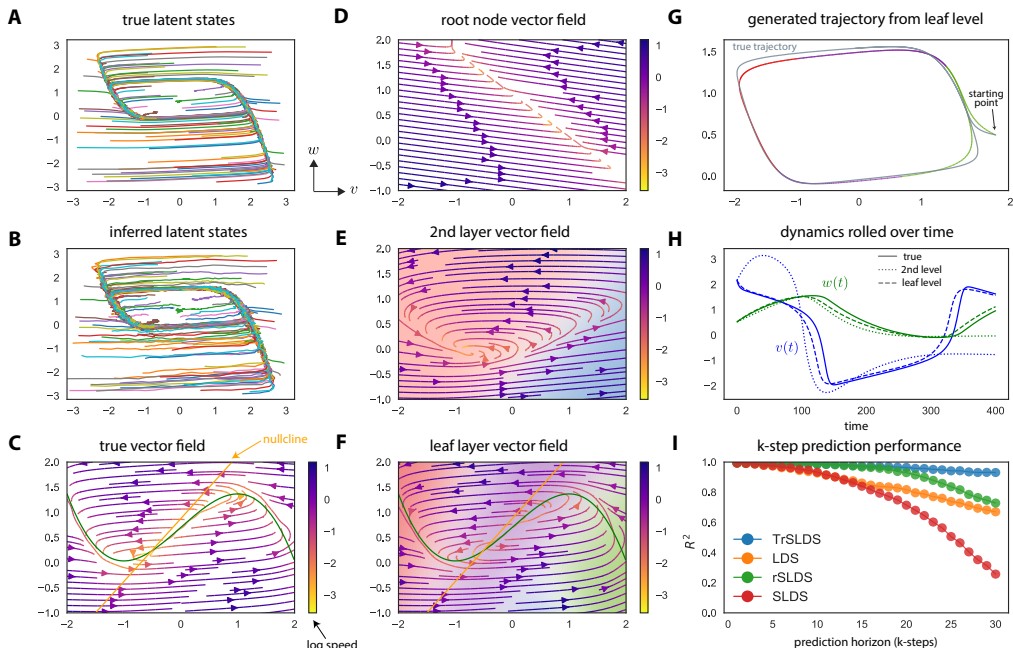

Figure 2: TrSLDS applied to model the FitzHugh-Nagumo nonlinear oscillator. **(a)** The model was trained on 100 trajectories with random starting points. **(b)** The model can infer the latent trajectories. **(c)** The true vector field of FHN is shown where color of the arrow represents log-speed. The two nullclines are plotted in yellow and green. **(d-f)** The vector fields display the multi-scale view learned from the model where color of the arrows dictate log-speed The background color showcases the hierarchical partitioning learned by the model where the darker the color is, the higher the probability of ending up in that discrete state. As we go deeper in the tree, the resolution increases which is evident from the vector fields. **(g)** A deterministic trajectory from the leaf nodes (colored by most likely leaf node) with affine transformation onto a trajectory FHN (gray). **(h)** Plotting $w$ and $v$ over time, we see that the second level captures some of the oscillations but ultimately converges to a fixed point. The model learned by the leaf nodes captures the limit cycle accurately. **(i)** Performances compared for multi-step prediction. We see that TrSLDS outperforms rSLDS.

## 5.1 FITZHUGH-NAGUMO

The FitzHugh-Nagumo (FHN) model is a 2-dimensional reduction of the Hodgkin-Huxley model which is completely described by the following system of differential equations (Izhikevich, 2007):

$$\dot{v} = v - \frac{v^3}{3} - w + I_{ext}, \qquad \tau\dot{w} = v + a - bw. \tag{24}$$

We set the parameters to $a = 0.7$, $b = 0.8$, $\tau = 12.5$, and $I_{ext} \sim \mathcal{N}(0.7, 0.04)$. We trained our model with 100 trajectories where the starting points were sampled uniformly from $[-3, 3]^2$. Each of the trajectories consisted of 430 time points, where the last 30 time points of the trajectories were used for testing. The observation model is linear and Gaussian where $C = \begin{pmatrix} 2 & 0 \\ 0 & -2 \end{pmatrix}$, $d = [0.5, 0.5]$ and $S = 0.01\mathbb{I}_2$ where $\mathbb{I}_n$ is an identity matrix of dimension n. We set the number of leaf nodes to be 4 and ran Gibbs for 1,000 samples; the last 50 samples were kept and we choose the sample that produced the highest log likelihood to produce Fig. 2 where the vector fields were produced using the mode of the conditional posteriors of the dynamics.

To quantitatively measure the predictive power of TrSLDS, we compute the $k$-step predictive mean squared error, $\text{MSE}_k$, and its normalized version, $R_k^2$, on a test set where $\text{MSE}_k$ and $R_k^2$ are defined as

$$\text{MSE}_k = \frac{1}{T-k}\sum_{t=0}^{T-k}\|y_{t+k} - \hat{y}_{t+k}\|_2^2, \qquad R_k^2 = 1 - \frac{(T-k)\text{MSE}_k}{\sum_{t=0}^{T-k}\|y_{t+k} - \bar{y}\|_2^2}, \tag{25}$$

where $\bar{y}$ is the average of a trial and $\hat{y}_{t+k}$ is the prediction at time $t+k$ which is obtained by (i) using the the samples produced by the sampler to obtain an estimate of $\hat{x}_T$ given $y_{1:T}$, (ii) propagate $\hat{x}_T$ for $k$ time steps forward to obtain $\hat{x}_{t+k}$ and then (iii) obtain $\hat{y}_{t+k}$. We compare the model to LDS, SLDS and rSLDS for $k = 1, \ldots, 30$ over the last 30 time steps for all 100 trajectories (Fig. 2I).

## 5.2    LORENZ ATTRACTOR

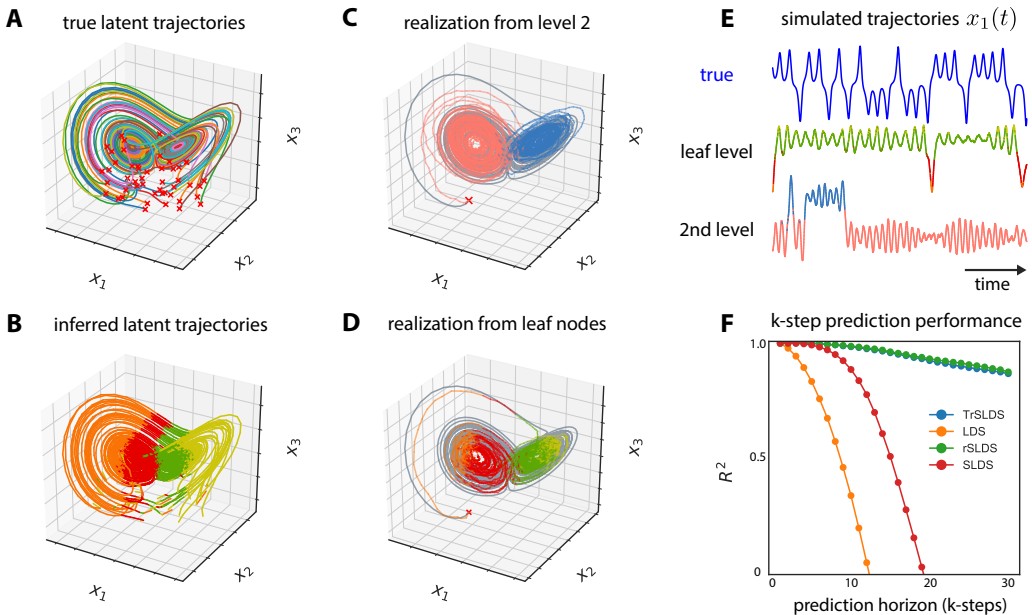

Figure 3: **(a)** The 50 trajectories used to train the model are plotted where the red "x" displays the starting point of the trajectory. **(b)** The inferred latent states are shown, colored by their discrete latent state. **(c)** We see that the second layer approximates the Lorenz attractor with 2 ellipsoids. A trajectory from the Lorenz attractor starting at the same initial point is shown for comparison. **(d)** Going one level lower in the tree, we see that in order to capture the nuances of the dynamics, each of the ellipsoids must be split in half. A trajectory from the Lorenz attractor is shown for comparison. **(e)** Plotting the dynamics, it is evident that the leaf nodes improve on it's parent's approximation. **(f)** The $R_k^2$ demonstrates the predictive power of TrSLDS.

Lorenz attractors are chaotic systems whose nonlinear dynamics are defined by,

$$\dot{x}_1 = \sigma\,(x_2 - x_1)\,, \quad \dot{x}_2 = x_1(\rho - x_3) - x_2\,, \quad \dot{x}_3 = x_1 x_2 - \beta x_3.$$

The parameters were set to $\sigma = 10$, $\rho = 28$ and $\beta = 8/3$. The data consisted of 50 trajectories, each of length of 230 where the first 200 time points are used for training and the last 30 are used for testing. The observation model was a projection onto 10 dimensional space with Gaussian noise. We set the number of leaf nodes to be 4 and ran Gibbs for 1,000 samples; the last 50 samples were kept and we choose the sample that produced the highest log-likelihood to produce Fig. 3.

The butterfly shape of the Lorenz attractor lends itself to being roughly approximated by two 2-dimensional ellipsoids; this is exactly what TrSLDS learns in the second level of the tree. As is evident from Fig. 5B, the two ellipsoids don't capture the nuances of the dynamics. Thus, the model partitions each of the ellipsoids to obtain a finer description. We can see that embedding the system with a hierarchical tree-structured prior allows for the children to build off its parent's approximations.

## 5.3    NEURAL DATA

To validate the model and inference procedure, we used the neural spike train data recorded from the primary visual cortex of an anesthetized macaque monkey collected by Graf et al. (2011). The dataset is composed of short trials where the monkey viewed periodic temporal pattern of motions

of 72 orientations, each repeated 50 times. Dimensionality reduction of the dataset showed that for each orientation of the drifting grating stimulus, the neural response oscillates over time, but in a stimulus dependent geometry captured in 3-dimensions (Zhao & Park, 2017). We used 50 trials each from a subset of 4 stimulus orientations grouped in two (140 and 150 degrees vs. 230 and 240 degrees) where each trial contained 140 neurons. Out of the 140 neurons, we selected 63 well-tuned neurons. The spike trains were binarized with a 10 ms window for Bernoulli observation model and we truncated the onset and offset neural responses, resulting in 111 time bins per trial.

We fit TrSLDS with $K = 4$ leaf nodes and 3-dimensional continuous latent space; the sampler was run for 500 samples where the last sample was used to produce the results shown in Fig. 4. To obtain an initial estimate for $x_{0:T}$, we smoothed the spike trains using a Gaussian kernel and performed probabilistic PCA on the smoothed spike trains.

From Fig. 4, it is evident that TrSLDS has learned a multi-scale view as expected. It is able to correctly distinguish between the two groups of orientations by assigning them to two different subtrees (green-yellow vs. red-orange). The leaf nodes of each subtree refines the periodic orbit further. From Fig. 4, we can see that TrSLDS also learns two limit cycles that are separated.

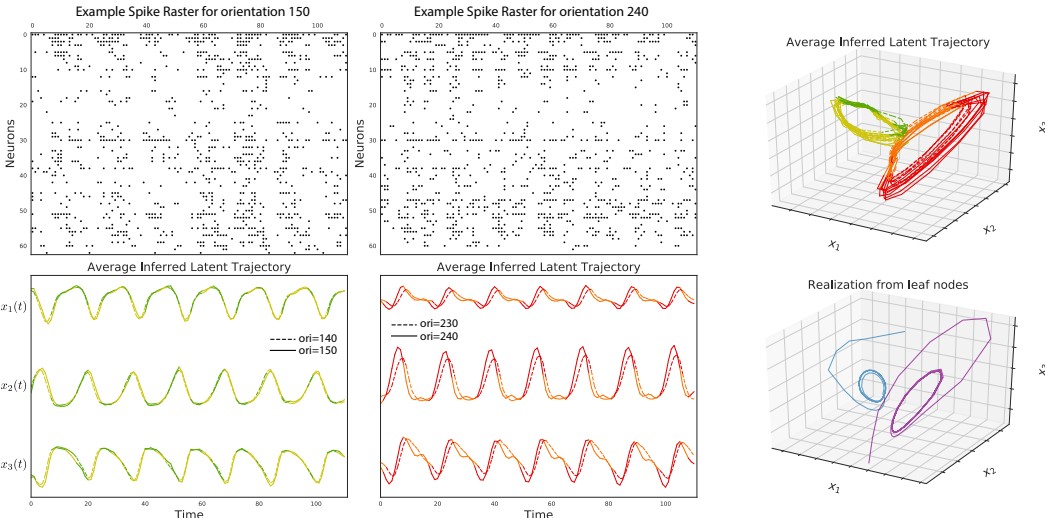

Figure 4: Modeling primary visual cortex spike trains. **(top)** Example spike raster plots in response to a drifting grating of orientations 150 and 240 degrees. Our data consisted of 200 such trials. **(bottom)** The average inferred latent trajectories over time for orientations 140 and 150 degrees colored by the most likely discrete latent state. **(right top)** Same plotted in space. The model is able to separate the limit cycles for each orientation group (green-yellow vs. red-orange) and refine them further with the leaf nodes. **(right bottom)** Two model generated predictive trajectories showing two stable limit cycles that resemble the two periodic orbits.

## 6    CONCLUSION

In this paper, we propose tree-structured recurrent switching linear dynamical systems (TrSLDS) which is an extension of rSLDS (Linderman et al., 2017). The system relies on the use of tree-structured stick-breaking to partition the space. The tree-structured stick-breaking paradigm naturally lends itself to imposing a hierarchical prior on the dynamics that respects the tree structure. This tree-structured prior allows for a multi-scale view of the system where one can query at different levels of the tree to see different scales of the resolution. We also developed a fully Bayesian sampler, which leverages the Pólya-Gamma augmentation, to learn the parameters of the model and infer latent states. The two synthetic experiments show that TrSLDS can recover a multi-scale view of the system, where the resolution of the system increase as we delve deeper into the tree. The analysis on the real neural data verifies that TrSLDS can find a multi-scale structure.

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

# A   PROOF OF THEOREM 1

*Proof.* Let $\mathcal{T}$ be a balanced binary tree with $K$ leaf nodes. To show that the models are equal, it suffices to show the equivalence of the likelihood and the prior between models. For compactness, we drop the affine term and the $\text{vec}(\cdot)$ operator. The likelihood of TrSLDS is

$$p(x_{1:T}|z_{1:T}, \Theta) = \prod_{t=1}^{T} \mathcal{N}(x_t|x_{t-1} + A_{z_t} x_{t-1}, Q_{z_t}), \tag{26}$$

and the likelihood of the residual model is

$$p(x_{1:T}|z_{1:T}, \tilde{\Theta}) = \prod_{t=1}^{T} \mathcal{N}\left(x_t|x_{t-1} + \bar{A}_{z_t} x_{t-1}, \tilde{Q}_{z_t}\right). \tag{27}$$

where $\bar{A}_{z_t}$ is defined in eq. (16). Substituting $A_{z_t} = \sum_{j \in \text{path}(z_t)} \tilde{A}_j$ into eq. (27) equates the likelihoods. All that is left to do is to show the equality of the priors.

We can express $A_n = \sum_{j \in \text{path}(n)} \tilde{A}_j$ recursively

$$A_n = \tilde{A}_n + A_{\text{par}(n)}. \tag{28}$$

Plugging eq. (28) into $\ln p(A_n|A_{\text{par}(n)})$

$$\ln p(A_n|A_{\text{par}(n)}) = -\frac{1}{2}\left(A_n - A_{\text{par}(n)}\right)^T \Sigma_n^{-1}\left(A_n - A_{\text{par}(n)}\right) + C \tag{29}$$

$$= -\frac{1}{2}\left(\tilde{A}_n + A_{\text{par}(n)} - A_{\text{par}(n)}\right)^T \Sigma_n^{-1}\left(\tilde{A}_n + A_{\text{par}(n)} - A_{\text{par}(n)}\right) + C \tag{30}$$

$$= -\frac{1}{2}\tilde{A}_n^T \Sigma_n^{-1} \tilde{A}_n + C \tag{31}$$

$$= -\frac{1}{2}\tilde{A}_n^T \left(\lambda^{\text{depth}(n)}\Sigma_\epsilon\right)^{-1} \tilde{A}_n + C \tag{32}$$

where C is a constant. Because $\Sigma_\epsilon = \tilde{\Sigma}_\epsilon$ and $\lambda = \tilde{\lambda}$, eq. (32) is equivalent to the kernel of $p(\tilde{A}_n)$ implying that the priors are equal. Since this is true $\forall n \in \mathcal{T}$, the joint distributions of the two models are the same. $\square$

# B   DETAILS ON BAYESIAN INFERENCE

## B.1   HANDLING BERNOULLI OBSERVATIONS

Suppose the observation of the system at time $t$ follows

$$p(y_t|x_t, \Psi) = \prod_{n=1}^{N} \text{Bern}(\sigma(\upsilon_{n,t})) = \prod_{n=1}^{N} \frac{(e^{\upsilon_{n,t}})^{y_{n,t}}}{1 + e^{\upsilon_{n,t}}}, \tag{33}$$

$$\upsilon_{n,t} = c_n^T x_t + d_n, \tag{34}$$

where $c_n \in \mathbb{R}^{d_x}$, $d_n \in \mathbb{R}$. Equation 33 is of the same form as the left hand side of eq. (20), thus it is amenable to PG augmentation. We introduce PG axillary variables $\eta_{n,t}$. Conditioning on $\eta_{1:N}$, eq. (33) becomes

$$p(y_t|x_t, \eta_{1:N}) = \prod_{n=1}^{N} e^{-\frac{1}{2}(\eta_{n,t}\upsilon_{n,t} - 2\kappa_{n,t}\upsilon_{n,t})} \tag{35}$$

$$\propto \prod_{n=1}^{N} \mathcal{N}(c_n^T x_t + d_n|\kappa_{n,t}/\eta_{n,t}, 1/\eta_{n,t}) \tag{36}$$

$$= \mathcal{N}(Cx_t + D|H_t^{-1}\boldsymbol{\kappa}_t, H_t^{-1}) \tag{37}$$

where $H_t = \text{diag}([\eta_{1,t}, \ldots, \eta_{N,t}])$, $\boldsymbol{\kappa}_t = [\kappa_{1,t}, \ldots, \kappa_{N,t}]$ and $\kappa_{n,t} = y_{n,t} - \frac{1}{2}$.

The observation is now effectively Gaussian and can be incorporated into the message passing for $x_{1:T}$. The emission parameters are also conjugate with the augmented observation potential given a Matrix Normal prior. The conditional posterior on the axillary PG variables $\eta_{n,t}$ also follows a PG distribution i.e. $\eta_{n,t}|(c_n, d_n), x_t \sim \mathrm{PG}(1, v_{n,t})$. Note that this augmentation scheme can also work for negative binomial, binomial, and multinomial observations (Polson et al., 2013; Linderman et al., 2015).

## B.2 MESSAGE PASSING FOR $x_{1:T}$

Assuming that the observations, $y_{1:T}$, are linear and Gaussian, the posterior of the continuous latent states, $x_{0:T}$, conditioned on all the other variables is proportional to

$$\prod_{t=1}^{T} \psi(x_t, x_{t-1}, z_t)\psi(z_t, x_{t-1})\psi(x_t, y_t) \tag{38}$$

where $\psi(x_t, x_{t-1}, z_t)$ is the potential of the conditionally linear dynamics, $\psi(x_t, y_t)$ is the potential of the observation and $\psi(x_{t-1}, z_t)$ is the recurrence potential. $\psi(x_{t-1}, z_t)$ is a product of all the internal nodes traversed at time $t$

$$\psi(x_{t-1}, z_t) = \prod_{n \in \mathrm{path}(z_t) \setminus \mathcal{Z}} \psi_n(x_{t-1}, z_t). \tag{39}$$

If the potentials in eq. (38) were all linear and Gaussian, then we could efficiently sample from the posetrior of $x_{0:T}$ by passing messsages forward through Kalman Filtering and then sampling backwards; the prescence of the recurrence potentials prevent this because they aren't Gaussian. By augmenting the model with the PG r.v.'s, the recurrence potential at internal node $n$ becomes

$$\psi_n(x_{t-1}, z_t, w_{n,t-1}) = \mathcal{N}(R_n^T x_{t-1} + r_n | \kappa_{n,t-1}/\omega_{n,t-1}, 1/\omega_{n,t-1}) \tag{40}$$

which is effectively Gaussian , allowing for the use of the Kalman filter for message passing.

## C INITIALIZATION

We initialized the Gibbs sampler using the following initialization procedure: (i) probabilistic PCA was performed on the data, $y_{1:T}$ to initialize the emission parameters, $\{C, d\}$ and the continuous latent states, $x_{1:T}$. (ii) To initialize the dynamics of the nodes ,$\Theta$, and the hyperplanes, $\Gamma$, we propose greedily fitting the proposed model using MSE as the loss function. We first optimize over the root node

$$\underset{A_\epsilon, b_\epsilon}{\arg\min} \frac{1}{T} \sum_{t=0}^{T} \|x_{t+1} - x_t - A_\epsilon x_t - b_\epsilon\|_2^2, \tag{41}$$

and obtain $A_\epsilon^*, b_\epsilon^*$ (Note that $A_\epsilon^*, b_\epsilon^*$ can obtained in closed form by computing their corresponding OLS estimates). Fixing $A_\epsilon^*$ and $b_\epsilon^*$, we then optimize over the second level in the tree

$$\underset{A_1, b_1, A_2, b_2, R_\epsilon, r_\epsilon}{\arg\min} \frac{1}{T} \sum_{t=0}^{T} \|x_{t+1} - \sigma(v_\epsilon)\hat{x}_1 - \sigma(-v_\epsilon)\hat{x}_2\|, \tag{42}$$

$$\hat{x}_i = x_t + (A_\epsilon^* + A_i) x_t + (b_\epsilon^* + b_i), \tag{43}$$

$$v_\epsilon = R_\epsilon^T x_t + r_\epsilon. \tag{44}$$

This procedure would continue until we reach the leaf nodes of the tree. $\Theta^*$ and $\Gamma^*$ are then used to initialize the dynamics and the hyperplanes, respectively. In our simulations, we used stochastic gradient descent with momentum to perform the optimization. (iii) The discrete latent states, $z_{1:T}$, were initialized by performing hard classification using $\Gamma^*$ and the initial estimate of $x_{0:T}$.

## D DEALING WITH ROTATIONAL INVARIANCE

A well known problem with these types of model is it's susceptibility to rotational and scaling transformation, thus we can only learn the dynamics up to an affine transformation Erosheva & Curtis (2017). During Gibbs sampling the parameters will continuously rotate and scale, which can slow

down the mixing of the chains. One possible solution to the issue is if we constrained $C$ to have some special structure which would make the model identifiable; this would require sampling from manifolds which is usually inefficient. Similar to Geweke & Zhou (1996), we use the following procedure to prevent the samples from continuously rotating and scaling:

- Once we obtain a sample from the conditional posterior of the emission parameters $\{C, D\}$, we normalize the columns of $C$.

- RQ decomposition is performed on $C$ to obtain $U, O$ where $U \in \mathcal{R}^{d_y \times d_x}$ is an upper triangular matrix and $O \in \mathcal{R}^{d_x \times d_x}$ is an orthogonal matrix.

- We set $C = U$ and rotate all the parameters of the model using $O$.

# E    SCALABILITY AND COMPUTATIONAL COMPLEXITY OF THE INFERENCE

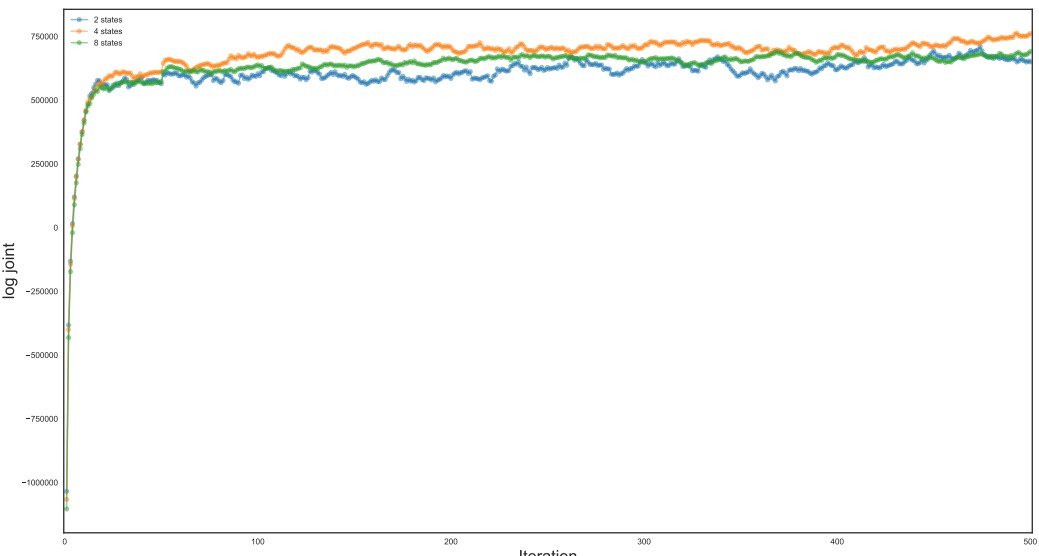

Figure 5: The logarithm of the joint density was computed for all the samples generated from the 3 TrSLDS and smoothed using a trailing moving average filter. The sampler seems to converge to a mode rather quickly for all the three instantiations of the TrSLDS.

The rSLDS and the TrSLDS share the same linear time complexity for sampling the discrete and continuous states, and both models learn K-1 hyperplanes to weakly partition the space. Specifically, both models incur: an $\mathcal{O}(TK)$ cost for sampling the discrete states, which increases to $\mathcal{O}(TK^2)$ if we allow Markovian dependencies between discrete states; an $\mathcal{O}(TD^3)$ cost (D is the continuous state dimension) for sampling the continuous states, just like in a linear dynamical system; and an $\mathcal{O}(KD^3)$ cost for sampling the hyperplanes. The only additional cost of the TrSLDS stems from the hierarchical prior on state dynamics. Unlike the rSLDS, we impose a tree-structured prior on the dynamics to encourage similar dynamics between nearby nodes in the tree. Rather than sampling K dynamics parameters, we need to sample 2K-1. Since they are all related via a tree-structured Gaussian graphical model, the cost of an exact sample is $\mathcal{O}(KD^3)$ just as in the rSLDS, with the only difference being a constant factor of about 2. Thus, we obtain a multi-scale view of the underlying system with a negligible effect on the computational complexity.

To see how the number of discrete latent states effects the convergence speed of the Gibbs sampler, we fit 3 TrSLDS, with $K = 2, 4, 8$ respectively, to a Lorenz Attractor described in Sec. but used 250 trajectories to train the model as opposed to 50. To assess convergence, we plotted the logarithm of the joint density as a function of Gibbs samples. The results are shown Fig. 5.

## F    SYNTHETIC NASCAR®

We ran the TrSLDS on the synthetic NASCAR® example from (Linderman et al., 2017), where the underlying model is an rSLDS. We trained TrSLDS on 10,000 time points and ran Gibbs for 1,000 samples; the last sample was used to create Fig. 6 where the vector fields where created from the mode of the conditional posterior of the dynamics. Even though the partitions were created using sequential stick-breaking, TrSLDS was able to reconstruct the dynamics of the model.

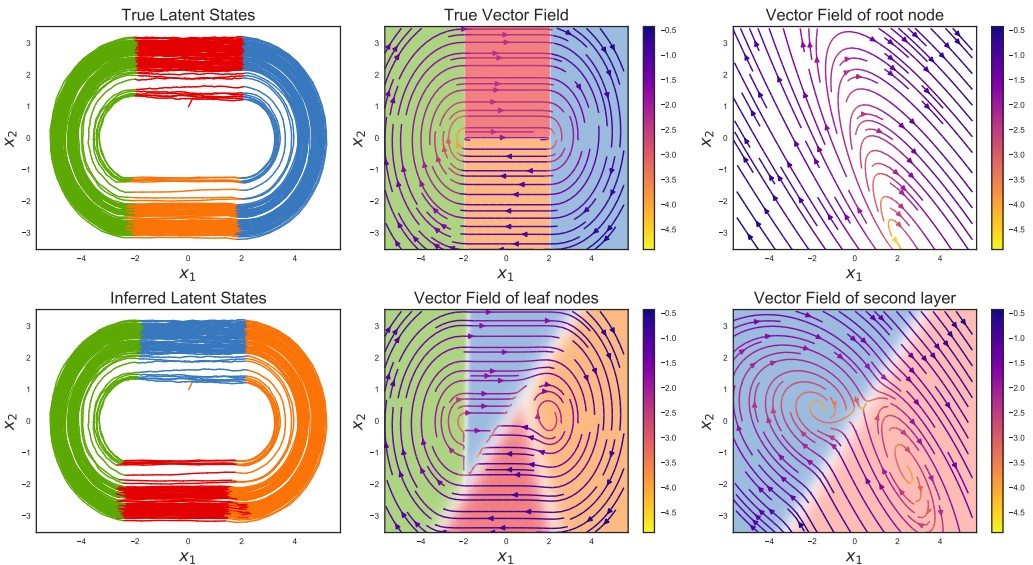

Figure 6: TrSLDS applied to the synthetic NASCAR® example. **(left top)** The true continuous latent states colored by their discrete state assignment. **(left bottom)** TrSLDS is able to infer the continuous and discrete latent states. **(middle top)** The dynamics were constructed such that the trajectories produced creates an oval track. **(middle bottom)** Although the partitioning is done through sequential stick-breaking, TrSLDS is still able to recover the dynamics. **(right)** We can see that TrSLDS can indeed recover a multi-scale view. The root nodes captures the rotation. The second level seperates the track into two rotations of different speeds.

## G    TREE SYNTHETIC NASCAR®

To check whether the sampler is mixing adequately, we test TrSLDS on a twist on the synthetic NASCAR® example where the underlying model is a TrSLDS. We also ran rSLDS on the example to highlight the limitations of seqeuntail stick-breaking. We trained both rSLDS and TrSLDS on 20,000 time points and Gibbs was ran for 1,000 samples; the last sample was used to create Fig. 7. To compare the precditive performance between the models, the $R_k^2$ was computed for rSLDS and TrSLDS.

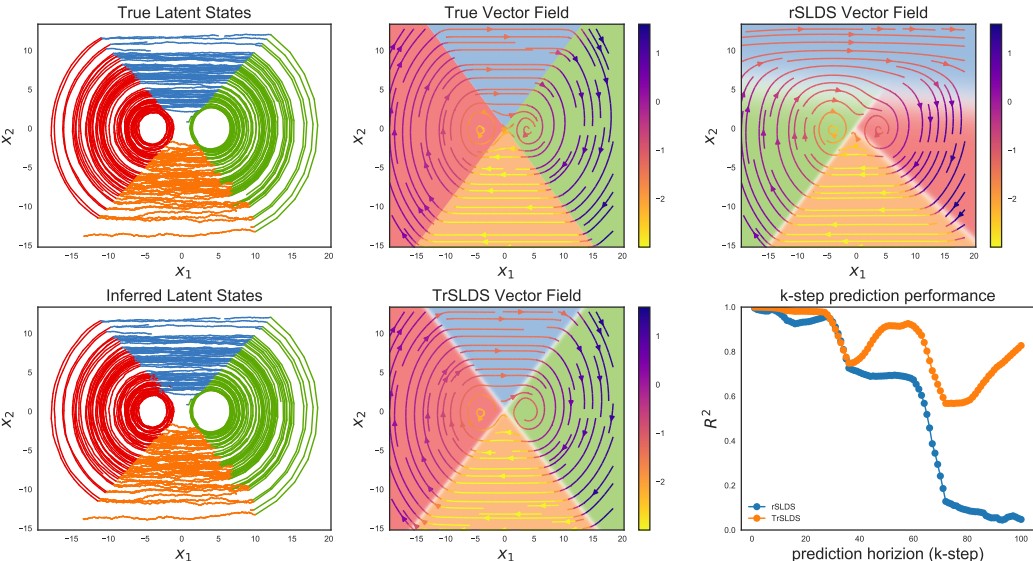

Figure 7: TrSLDS and rSLDS applied to the tree version of to the synthetic NASCAR®. **(left top)**The true continuous latent states colored by their discrete state assignment. **(left bottom)** TrSLDS is able to infer the continuous and discrete latent states. **(middle)** TrSLDS was able to learn the underlying dynamics and partitions, indiicating that the sampler is mixing well. **(right top)**Due to the sequential nature of rSLDS, the model can't adequately learn the dynamics of the model. **(right bottom)** We can see from the k-step $R^2$ that TrSLDS outperforms rSLDS.

