# OpenReview forum: "Tree-Structured Recurrent Switching Linear Dynamical Systems for Multi-Scale Modeling"
_ICLR.cc/2019/Conference_

### Official Review · AnonReviewer2 · 2018-11-03
**Interesting extension to switching linear dynamical systems; scalability concerns and issues with experiments.**

**Rating:** 6
**Confidence:** 4

**Review:**

The authors develop a tree structured extension to the recently proposed recurrent switching linear dynamical systems. Like switching linear dynamical systems (sLDS) the proposed models capture non-linear dynamics by switching between a collection of linear regimes. However, unlike SLDS, the transition between the regimes is a function of a latent tree as well as the preceding continuous latent state. Experiments on synthetic data as well as neural spike train data are presented to demonstrate the utility of the model.

The paper is clearly written and easy to read. The tree structured model (TrSLDS) is a sensible extension to rSLDS. While one wouldn’t expect TrSLDS to necessarily fit the data any better than rSLDS, the potential for recovering multi-scale, possibly more interpretable decompositions of the dynamic process is compelling.

While the authors do provide some evidence of being able to recover such multi-scale structures, overall the experiments are underwhelming and somewhat sloppy. First, to understand whether the sampler is mixing well, it would be nice to include an experiment where the true dynamics and the entire latent structure (including the discrete states) are known, and then to examine how well this ground-truth structure is recovered. Second, for the results presented in section 5, how many iterations was the sampler run for? In the figures, what is being visualized?, the last sample?, the MAP sample? or something else? I am not sure what to make of the real data experiment in section 5.3. Wouldn’t rSLDS produce nearly identical results? What is TrSLDS buying us in this scenario? Do the higher levels of the tree capture interesting low resolution dynamics that are not shown for some reason?

My other big concern is scalability. To use larger number of discrete states one would need deeper (or wider if the binary requirement is relaxed) trees. How well does the sampler scale with the number of discrete states? How long did the sampler take for the various 4-state results presented in the paper?

Minor:
a) There is a missing citation in the first para fo Section 5.
b) Details of message passing claimed to be in the supplement are missing.

============
There are interesting ideas in this paper. However, experimental section could better highlight the benefits afforded by the model and scalability concerns need to be addressed.

---

> ### Author Response · Authors · 2018-11-26
> **Response to Reviewer 2**
>
> We thank the reviewer for her/his insightful review. We note that tree-structured stick-breaking utilized by TrSLDS is a strict generalization of the sequential stick-breaking used in rSLDS. As stated in the response to AnonReviewer1 above, we can recover sequential stick-breaking from tree-structured stick-breaking by enforcing the left node at each level in the tree to be a leaf node. We have amended the manuscript to make this connection explicit.
>
> Since tree-structured stick-breaking is a strict generalization of sequential stick-breaking, the  expressive power of TrSLDS theoretically subsumes that of rSLDS. The question reduces to a comparison of tree structures; in our experiments, a comparison of right-branching trees to balanced binary trees. We emphasize this by including two new examples in the appendix in which the true dynamics and the entire latent structure are known; the first being the “synthetic Nascar” example used in Linderman et al. (2017), where the true model follows a right-branching tree, as in the standard rSLDS, to emphasize that we can effectively learn these dynamics with a tree-structured model. The second example is a twist on the synthetic Nascar where the underlying model is a TrSLDS and where we test both rSLDS and TrSLDS. In this example (Fig. 7), rSLDS fails due to the sequential nature of stick-breaking that cannot adequately capture the locally-linear dynamics.
>
> >>Experiment Section
> We thank the reviewer for pointing out the missing information in the experiments section and have amended the manuscript with corrections. As stated above, we have included two more examples in the appendix to highlight not only the expressive power of TrSLDS, but also to show that the sampler is indeed mixing well. Concerning the real data experiment, we have amended the manuscript with more results from the analysis. The orientations were chosen to resemble a tree where orientations 140 and 150 have the same parent; the same is true for orientations 230 and 240. Thanks to the multi-scale nature of TrSLDS, the method is able to learn this relation and assigns the two groups to different subtrees. It then refines the dynamics by focusing on each of these two groups separately.
>
> >>Scalability
> The computational complexity of TrSLDS is of the same order as rSLDS; for specifics please refer to our response to AnonReviewer 1.
>
> To address the concerns regarding the samplers mixing speed as a function of number of discrete states, we fit a TrSLDS with K = 2, 4, 8 discrete states, keeping the amount of data used to train the model fixed, and plotted the log joint density as function of samples and included it in the appendix. From the plots, the sampler seems to converge to a  mode of the posterior after about 150-250 samples for each of the various numbers of discrete states. Due to the nature of Gibbs sampling, we are limited to batch updates to each of the conditional posteriors. While scalability has not been an issue in our experiments,  we will explore stochastic variational approaches in future work.
>
> >>Minor
> We thank the reviewer for pointing out these minor mistakes and have corrected them in the amended manuscript.

---

> > ### Comment · AnonReviewer2 · 2018-11-27
> > **Updates sufficiently address my concerns**
> >
> > Thank you for the response and the significant updates to the paper. The rebuttal sufficiently addresses most of my concerns. Although, I still have some concerns about scalability, they are not a showstopper for me and I am willing to support the acceptance of this revised paper.

---

### Official Review · AnonReviewer1 · 2018-11-05
**This is an interesting paper that seems to make a non-trivial contribution. The paper is however not positioned against existing work in hierarchical SLDS and is also somewhat sloppy in the experiment.  I currently give this paper a borderline rating but would increase the scores if my concerns above are addressed satisfactorily.**

**Rating:** 7
**Confidence:** 2

**Review:**

PAPER SUMMARY:

This paper introduces a probabilistic generative framework to model linear dynamical systems at multiple levels of resolution, where the entire complex, nonlinear dynamics is approximated via a hierarchy of local regimes of linear dynamics -- the global dynamic is then characterized as a switching process that switches between linear regimes in the hierarchy.

NOVELTY & SIGNIFICANCE:

The key contributions of this paper are (a) the use of tree-structured stick breaking to partition the entire dynamic space into a hierarchy of linear regimes; (b) the design of a hierarchical prior that is compatible to the tree structure; and (c) the developed Bayesian inference framework for it in Section 4.

By exploiting the tree-structured stick breaking process (Adams et al., 2010), the proposed framework is able to partition the entire dynamic space into a hierarchy of switching linear regimes.

This allows the dynamic to be queried at multiple levels of resolution. This appears to be the key difference between the proposed framework and the previous work of (Linderman et al., 2017) on recurrent switching dynamical systems that partition the dynamic space sequentially at the same level of resolution.

This seems like a non-trivial extension to the previous work of (Linderman et al., 2017) & I tend to consider this a novel contribution. That said, the paper was also not positioned against existing literature on hierarchical switching linear dynamic systems (see below) & I find it hard to evaluate the significance of the proposed framework (which explains the borderline rating)

"A Hierarchical Switching Linear Dynamical System applied to the detection of sepsis in neonatal condition monitoring", Ioan Stanculescu, Christopher K. I. Williams and Yvonne Freer. In Proceeding of the 30th Conference on Uncertainty in AI (UAI-14), pages 752-761

Could the authors please discuss the differences between the proposed work & (at least) the above?

TECHNICAL SOUNDNESS:

The technical exposition makes sense to me. Please also discuss the processing complexity of the resulting TrSLDS framework. In exchange for the improved performance, how much slower TrSLDS is as compared
to rSLDS? I am interested to see this demonstrated in the empirical studies.

CLARITY:

The paper is clearly written.

EMPIRICAL RESULTS:

The experiments look interesting and are very extensive on both test domains. However, I do not understand why the authors decided not to compare with rSLDS using its benchmark?

I find this somewhat sloppy and hope the authors would clarify this too.

****

Post-rebuttal update: The authors have made significant revision to their work, which sufficiently addressed all my concerns. I have upgraded my score accordingly and I am willing to support the acceptance of this paper.

---

> ### Author Response · Authors · 2018-11-26
> **Response to Reviewer 1**
>
> We thank the reviewer for her/his insightful review and for bringing up the prior work done by Stanculescu et al. 2014. In Stanculescu et al. 2014., they propose adding a layer to factorized SLDS where the top-level discrete latent variables determine the conditional distribution of z_t, with no dependence on x_{t-1}. While the tree-structured stick-breaking used in TrSLDS is also a hierarchy of discrete latent variables, the model proposed in Stanculescu et al. 2014., has no hierarchy of dynamics, preventing it from obtaining a multi-scale view of the dynamics. Stanculescu et al. (2014) also reference preceding work on hierarchical SLDS by Zoeter & Heskes (2003), the only example they found in the literature. In Zoeter & Heskes (2003), the authors construct a tree of SLDSs where an SLDS with K possible discrete states is first fit. An SLDS with M discrete states is then fit to each of the K clusters of points. This process continues iteratively, building a hierarchical collection of SLDSs that allow for a multi-scale, low-dimensional representation of the observed data. While similar in spirit to TrSLDS, there are key differences between the two models.
> First, it is through the tree-structured prior that TrSLDS obtains a multi-scale view of the dynamics, thus we only need to fit one instantiation of TrSLDS; in contrast, they fit a separate SLDS for each node in the tree, which is computationally expensive. There is also no explicit probabilistic connection between the dynamics of a parent and child in Zoeter & Heskes (2003). We also note that TrSLDS aims to learn a multi-scale view of the dynamics while Zoeter & Heskes (2003) focuses on smoothing, that is, they aim to learn a multi-scale view of the latent states corresponding to data but not suitable for forecasting. We have amended the manuscript to include a section discussing prior and related work.
>
> >>Technical Soundness
> The rSLDS and the TrSLDS share the same linear time complexity for sampling the discrete and continuous states, and both models learn K-1 hyperplanes to weakly partition the space. Specifically, both models incur: an O(TK) cost for sampling the discrete states, which increases to O(TK^2) if we allow Markovian dependencies between discrete states; an O(TD^3) cost (D is the continuous state dimension) for sampling the continuous states, just like in a linear dynamical system; and an O(KD^3) cost for sampling the hyperplanes. The only additional cost of the TrSLDS stems from the hierarchical prior on state dynamics.  Unlike the rSLDS, we impose a tree-structured prior on the dynamics to encourage similar dynamics between nearby nodes in the tree.  Rather than sampling K dynamics parameters, we need to sample 2K-1.  Since they are all related via a tree-structured Gaussian graphical model, the cost of an exact sample is O(KD^3) just as in the rSLDS, with the only difference being a constant factor of about 2. Thus, we obtain a multi-scale view of the underlying system with a negligible effect on the computational complexity. We have amended the manuscript to make this clear.
>
> We also note that tree-structured stick-breaking utilized by TrSLDS is a strict generalization of the sequential stick-breaking used by rSLDS. We can recover sequential stick-breaking from tree-structured stick-breaking by enforcing the left node at each level in the tree to be a leaf node. Our experiments only considered balanced binary trees for simplicity, but an interesting avenue of future work is to learn the tree structure, perhaps through additional MCMC. Learning such discrete representations is highly non-trivial and demands further investigation outside this submission.  We have amended the manuscript to make this connection explicit.
>
> >>Empirical Results
> The Lorenz attractor in experiment 2 was also used in as a benchmark for the rSLDS (Linderman et al, 2017; Fig. 4). The only difference is that Linderman et al generated binary observations with a Bernoulli GLM emission model. For completeness, we ran TrSLDS on the synthetic nascar example used to test rSLDS in Linderman et al. (2017) to see if we could recover the dynamics and the discrete latent state assignments and included the results in the appendix. We also note that we included another example in the appendix where the data generated from an alternative version of the synthetic nascar example from Linderman et al. (2017) where the underlying model is a TrSLDS and compared both TrSLDS and rSLDS.

---

> > ### Comment · AnonReviewer1 · 2018-12-06
> > **Response to Authors' Revision**
> >
> > Thank you for the significant updates and detailed clarification. The revision has sufficiently addressed all my concerns. I have upgraded my score and am willing to support the acceptance of this paper.

---

### Official Review · AnonReviewer3 · 2018-11-06
**a tree extension to previous work rSLDS**

**Rating:** 7
**Confidence:** 2

**Review:**

This paper introduces a probabilistic model to model nonlinear dynamic systems with multiple granularities. The nonlinearity is achieved by using multiple local linear approximations. The method is an extension to rSLDS (recurrent switching linear dynamical systems),  which in turn is an extension to SLDS.

Pros:
1. Introducing the tree structure is a neat way of extending the existing rSLDS model to multiscale scenarios.
2. The paper is written clearly. The background is well illustrated and the idea rises naturally from there. The paper is also solid in the part describing the model.
Con:
1. from the rSLDS paper (https://arxiv.org/pdf/1610.08466.pdf), the authors there was experimenting with some settings similar to those used in this paper. However, I am not able to find some explicit comparison between the TrSLDS and rSLDS in this work. I think it should be needed since TrSLDS itself is derived out from rSLDS, it would be good to show explicitly the advantage of the new model.

---

> ### Author Response · Authors · 2018-11-26
> **Response to Reviewer 3**
>
> We thank the reviewer for her/his insightful review. We note that we added a section to the Appendix comparing the computational complexity of TrSLDS and rSLDS in which it states that the computational complexity of TrSLDS is of the same order as rSLDS; for specifics please refer to our response to AnonReviewer 1. Thus, we obtain a multi-scale view of the underlying system with a negligible effect on the computational complexity.  We also note that tree-structured stick-breaking utilized by TrSLDS is a strict generalization of the sequential stick-breaking used by rSLDS; we have amended the manuscript to make this connection explicit.
>
> For both synthetic experiments, the predictive power of TrSLDS and rSLDS (as well as SLDS and LDS)  was compared using k-step R^2 (Figs 2 & 3). In both synthetic experiments, the predictive power of TrSLDS is at least as much rSLDS.
>
> To better highlight the differences between TrSLDS and rSLDS, we added two more examples in the appendix. The first example is the synthetic NASCAR from (https://arxiv.org/pdf/1610.08466.pdf) where the underlying model is indeed an rSLDS (Fig. 6). The space is partitioned into 4 sections using sequential stick-breaking, where the trajectories trace out oval tracks similar to a NASCAR track. TrSLDS was fit to see if it could recover the dynamics even though it relies on tree-structured stick-breaking. From Fig. 6, it is evident that TrSLDS can recover the dynamics and obtain a multi-scale view. The second example is a twist on the TrSLDS, where the underlying model is a TrSLDS i.e. the space is partitioned using tree-structured stick-breaking (Fig. 7). We ran TrSLDS and rSLDS and compared their predictive performance using k-step R^2. From Fig. 7, we can see that rSLDS could not adequately learn the vector field due to its reliance on sequential stick-breaking. This provides empirical evidence that the expressive power of TrSLDS subsumes that of rSLDS which was stated in our response to AnonReviewer 1.

---

> > ### Comment · AnonReviewer3 · 2018-12-10
> > **Response to authors' rebuttal**
> >
> > Thanks to the authors for the detailed and sufficient contents added to the appendix. I am satisfied with the new proof provided by the author and am willing to support it to be accepted. My score to the paper is also updated accordingly.

---

### Author Response · Authors · 2018-11-26
**Summary of revisions**

We thank all the reviewers for their suggestions and have amended the manuscript accordingly. Here is a summary of the changes:
1) Added a paragraph discussing prior work on hierarchical extensions of SLDS.
2) Added section describing the Polya-Gamma data augmentation scheme.
3) Redid the experiments to better highlight the multi-scale nature of our algorithm. (Figs 2, 3, 4)
4) Added a section in the appendix describing how to handle Bernoulli observations using Polya-Gamma data augmentation scheme.
5) Added a section in the appendix providing details on the message-passing used in the sampling.
6) To better highlight the differences between rSLDS and TrSLDS, two new experiments have been added to the appendix including a benchmark experiment from the original rSLDS paper. (Figs. 6 & 7)
7) Added a section in the appendix discussing the computational complexity of fitting the model. We also show empirically how the time till convergence of the MCMC sampler changes as a function of discrete latent states by fitting three TrSLDS of varying number of leaf nodes and plot the log of the joint density. (Fig. 5)

---

### Meta-Review · Area_Chair1 · 2018-12-16
**Good paper on modeling nonlinear dynamical system.**

**Confidence:** 5
**Recommendation:** Accept (Poster)

**Metareview:**

This paper presents a recurrent tree-structured linear dynamical system to model the dynamics of a complex nonlinear dynamical system. All reviewers agree that the paper is interesting and useful, and is likely to have an impact in the community. Some of the doubts that reviewers had were resolved after the rebuttal period.

Overall, this is a good paper, and I recommend an acceptance.